# Post-Coronavirus Disease 2019 Pandemic Antimicrobial Resistance

**DOI:** 10.3390/antibiotics13030233

**Published:** 2024-02-29

**Authors:** Lucia Boccabella, Elena Gialluca Palma, Ludovico Abenavoli, Giuseppe Guido Maria Scarlata, Mariavirginia Boni, Gianluca Ianiro, Pierangelo Santori, Jan F. Tack, Emidio Scarpellini

**Affiliations:** 1Internal Medicine Unit, Madonna del Soccorso General Hospital, Via Luciano Manara 7, 63074 San Benedetto del Tronto, Italy; lucia.boccabella@sanita.marche.it (L.B.); pierangelo.santori@sanita.marche.it (P.S.); 2Internal Medicine Clinics, Riuniti University Hospital, Polytechnics University of Marche, 60121 Ancona, Italy; elena.giallucapalma@univpm.it; 3Department of Health Sciences, University “Magna Graecia”, 88100 Catanzaro, Italy; l.abenavoli@unicz.it (L.A.); giuseppeguidomaria.scarlata@unicz.it (G.G.M.S.); 4Vascular Medicine Unit, “C. and G. Mazzoni” General Hospital, 63076 Ascoli Piceno, Italy; mariavirginia.boni@sanita.marche.it; 5Gastroenterology Unit, Fondazione Policlinico Gemelli, Catholic University of Sacred Heart, 00168 Rome, Italy; gianluca.ianiro@hotmail.it; 6Translational Research in GastroIntestinal Disorders (T.A.R.G.I.D.), Gasthuisberg University Hospital, KU Leuven, Herestraat 49, 3000 Leuven, Belgium; jan.tack@med.kuleuven.be

**Keywords:** antibiotics, antimicrobial resistance, severe acute respiratory syndrome coronavirus 2 (SARS-CoV2), COVID-19 pandemic, personal protective equipment

## Abstract

Background and aim: Antimicrobial resistance (AMR) is a chronic issue of our Westernized society, mainly because of the uncontrolled and improper use of antimicrobials. The coronavirus disease 2019 (COVID-19) pandemic has triggered and expanded AMR diffusion all over the world, and its clinical and therapeutic features have changed. Thus, we aimed to review evidence from the literature on the definition and causative agents of AMR in the frame of the COVID-19 post-pandemic era. Methods: We conducted a search on PubMed and Medline for original articles, reviews, meta-analyses, and case series using the following keywords, their acronyms, and their associations: antibiotics, antimicrobial resistance, severe acute respiratory syndrome coronavirus 2 (SARS-CoV2), COVID-19 pandemic, personal protective equipment. Results: AMR had a significant rise in incidence both in in-hospital and outpatient populations (ranging from 5 up to 50%) worldwide, but with a variegated profile according to the germ and microorganism considered. Not only bacteria but also fungi have developed more frequent and diffuse AMR. These findings are explained by the increased use and misuse of antibiotics and preventive measures during the first waves of the SARS-CoV2 pandemic, especially in hospitalized patients. Subsequently, the reduction in and end of the lockdown and the use of personal protective equipment have allowed for the indiscriminate circulation of resistant microorganisms from low-income countries to the rest of the world with the emergence of new multi- and polyresistant organisms. However, there is not a clear association between COVID-19 and AMR changes in the post-pandemic period. Conclusions: AMR in some microorganisms has significantly increased and changed its characteristics during and after the end of the pandemic phase of COVID-19. An integrated supranational monitoring approach to this challenge is warranted in the years to come. In detail, a rational, personalized, and regulated use of antibiotics and antimicrobials is needed.

## 1. Introduction

Antimicrobial resistance (AMR) is understood as the ability of microorganisms to resist to antimicrobial treatments. This type of resistance has been defined by experts as the “slow pandemic” to emphasize its insidious and slow proliferation within our society and its potential danger to the economy and global health [1]. In fact, it is estimated that infections due to resistant pathogens cause approximately 700,000 deaths every year worldwide. This number is estimated to increase to 10 million deaths per year worldwide by 2050 [2]. Interestingly, AMR-related mortality in Italy accounts for almost 7000 deaths per year [3].

In recent years, this health emergency has been significantly amplified by another issue: the coronavirus disease 2019 (COVID-19) pandemic which caused more than 5 million deaths, infecting approximately 270 million people all over the world. Perhaps, the severe acute respiratory syndrome coronavirus 2 (SARS-CoV2) pandemic has entered its endemic phase and will stay with us for many years to come [4].

The SARS-CoV2 pandemic caused a massive wave of contagion, impacting our healthcare systems. The latter were unprepared in terms of both supplies of personal protective equipment (PPE) or individual protection devices (DPIs) and shortages in medications. Millions of antibiotics prescriptions were inappropriately and injudiciously prescribed within the drama of treating COVID-19 patients, and this inappropriate use has contributed to the emergence of newer antibiotic and antimicrobial resistance within the global population in a very complex fashion [4].

The effect of COVID-19 pandemic antimicrobial resistance is currently estimated to kill approximately 700,000 people per year in the general population [5]. This is a growing factor that must be pooled to the COVID-19 deaths per year to come, estimated to remain constant. For example, newer bacterial and fungal infections have colonized hospital-admitted patients (e.g., undergoing elective surgery, caesarean sections) and also outpatient clinics [6,7]. In this frame, the infection of certain non-immunocompromised subjects is also interesting. However, data from the literature are not uniform because of study design, patient type (from low intensity vs. intensive care clinic), and population differences (e.g., Western vs. Eastern world, high- vs. low-income countries). In fact, while there was an increased incidence and prevalence for AMR regarding certain Gram-negative and only a few Gram-positive bacteria, fungal infections showed some multi- and polyresistance profiles in immunocompromised patients. Indeed, fungal AMR diffusion worldwide still lacks evidence, although the COVID-19 pandemic seems to also be associated with infections in non-immunocompromised subjects [7,8]. Another aspect involved in AMR development in the post-pandemic time is the role of the potentially deleterious effects of the COVID-19 pandemic on AMR due to the deprioritization of infection prevention and control (IPC) and antimicrobial stewardship programs (ASPs) [4,8]. Finally, lockdown, reduced international travelling, and social distancing play a multifaceted role in AMR [4].

Therefore, altogether this evidence from the literature does not yet show a clear association between COVID-19 and the variegated AMR increase in the post-pandemic era.

Thus, the aim of this review was to define antimicrobial resistance, its incidence, and characteristics after the COVID-19 pandemic. We have reviewed evidence from the literature both on the short- and long-term effects of AMR, and also in developing countries [8].

## 2. Results and Discussion

### 2.1. Antimicrobial Resistance and Germs during the Pandemic

#### 2.1.1. The Concept of Antimicrobial Resistance

A bacterium is considered resistant to a specific antimicrobial agent when a recommended dose neither kills [9] nor effectively inhibits its multiplication [10,11].

The mechanism of bacterial AMR can be categorized as: (1) restrictive absorption of a drug; (2) alteration in the target of a drug; (3) drug inactivation; or (4) efflux pump. Restrictive drug absorption, drug inactivation, and active drug efflux are considered to be mechanisms of intrinsic resistance. On the other hand, acquired resistance mechanisms include the modification of the drug target in microbial cells, inactivation of the drug molecule, and modification of the drug efflux pump [12].

Gram-negative and Gram-positive bacteria display different mechanisms behind AMR, mainly because of their different cell structure. In fact, Gram-negative bacteria show AMR that is potentially mediated by any of the four main mechanisms [13]. In contrast, Gram-positive bacteria mainly resist antibiotics through restrictive drug absorption. In fact, they lack the external LPS membrane and have a limited capacity for specific drug efflux. Interestingly, Gram-negative bacteria use the LPS layer as a barrier to several drug molecules. Therefore, they have an “innate” resistance to different antimicrobials [14].

#### 2.1.2. Antibiotics Resistance and Germs: Gram-Negative Bacteria

The global increase in multidrug-resistant organisms (MDROs), such as carbapenem-resistant New Delhi metal-β-lactamase (NDM)-producing *Acinetobacter baumannii*, *Enterobacterales*, extended-spectrum β-lactamase (ESBL)-producing *Klebsiella pneumoniae*, and methicillin-resistant *Staphylococcus aureus* resistant (MRSA), has been particularly observed since the COVID-19 pandemic started [15,16]. In Italy, *Klebsiella pneumoniae* showed a small increase in incidence up to 29.5% (from 28.5%) of isolates in 2020. The isolates include the general population of patients. Similarly, *Pseudomonas aeruginosa* e *Acinetobacter* spp. showed an increasing incidence of 15.9 and 80.8% of isolates, respectively, vs. the previous year. In 2020, about 33,000 cases of deaths attributable to MDROs were recorded out of 633,000 registered cases [17], which account for non-COVID-19 patients.

However, according to a recent meta-analysis and systematic review of the literature, the impact of the COVID-19 pandemic on the increase in MDRO incidence/prevalence has been not statistically significant [18,19,20].

It is important to mention that the meta-analysis data reviewed have been extracted by authors according to the following methods: in studies providing complete numerator and denominator data, the incidence rate ratios (IRRs) were pooled using a GLMM random-effects meta-analysis, and the risk ratio (RRs) were pooled using Mantel–Haenszel random-effects meta-analysis with between-study variance estimated using the Paule–Mandel estimator. Moreover, incidence density (e.g., cases of resistant infections per 1000 patient days) was used to measure the change in AMR associated with COVID-19 or the proportion of isolates/infections, followed by incidence (e.g., cases per admission or discharges) and other measures (standardized infection ratio, point prevalence) [20,21].

In a study pooling all resistant Gram-negative organisms detected by the US Center for disease control (CDC), a non-statistically significant association was found between the COVID-19 pandemic and the incidence rate (IRR 1.64, 95% CI: 0.92–2.92, I2 = 93%, *n* = 14). Similar findings were reached for the percentage of resistant cases (RR 1.08, 95% CI: 0.91–1.29, I2 = 92%, *n* = 22). Failure to report enhanced Infection Prevention and Control (IPAC) and/or ASP was significantly associated with increased antimicrobial resistance in Gram-negatives (RR 1.11, 95% CI: 1.03–1.20, I2 = 88%, *n* = 5). On the other hand, there was no significant association with antimicrobial resistance in the studies considered (RR 0.80, 95% CI: 0.38–1.70, I2 = 90%, *n* = 17). Subsequently, testing for differences between subgroups showed no statistically significant differences between the presence and absence of enhanced IPAC/ASP interventions when evaluating changes in antimicrobial resistance (*p* = 0.4). In detail, there was no association between COVID-19 and the incidence of carbapenem- or multidrug-resistant *Acinetobacter* spp. (IRR 0.79, 95% CI: 0.30–2.07, I2 = 77%, *n* = 4) out of 325,847 patient days [18]. However, a small increase was found in the percentage of infections resistant to *Acinetobacter* spp. when comparing the pre- and post-COVID-19 pandemic periods (RR 1.02, 95% CI: 1.01–1.03, I2 = 0%, *n* = 2) [19]. It is interesting to look at the role of the dynamics of IPAC and ASP during the pandemic. According to an international survey by the Global Antimicrobial Resistance and Use Surveillance System (GLASS), the quality of many IPC measures improved in several countries during the pandemic. However, breaches in adherence to standard IPC practices were also reported in the survey [8]. The breaches can explain the differential associations with different germs’ AMR.

In analyzing the data on 1,609,923 patient days, no association was found between COVID-19 and the incidence of resistant *Pseudomonas* (IRR 1.10, 95% CI: 0.91–1.30, I2 = 0%, *n* = 4). Similarly, no association was found with the percentage of resistant cases (RR 1.02, 95% CI: 0.85–1.23, I2 = 58%, *n* = 6) [20,21]. The patients examined resemble the general population admitted to hospital and also those in tertiary care centers (e.g., Saint George Hospital University Medical Center, Beirut, Lebanon).

In a perspective French study evaluating 87,204 patients’ days of follow-up, there was an increased IRR associated with the COVID-19 pandemic (IRR 15.20, 95% CI: 4.90–47.14) in ESBL-producing (or third-generation cephalosporin-resistant) *Enterobacteria*. In detail, the ESBL-*E.coli* rates from clinical samples of patients in primary care and in nursing home residents were compared before and after the general lockdown in March 2020. However, the proportion of cases with an ESBL-producing organism was not affected by COVID-19 diagnosis (RR: 1.10, 95% CI: 0.91–1.33, I2 = 94%, *n* = 8) [22,23].

In a Brazilian 587,047-patient-day study, no significant change in the incidence of carbapenem-resistant *Enterobacteria* (CRE), namely, *E. coli* and *Klebsiella* spp., was found (IRR 2.05, 95% CI: 0. 77–5.44, I2 = 95%, *n* = 5). Accordingly, no increase in the proportion of CRE cases was identified (RR 1.10, 95% CI: 0.61–1.99, I2 = 88%, *n* = 6) [24]. Patients were compared with corresponding adult patients admitted to the ICU from April through June 2020 (namely, pandemic period) with the same period in 2019 (namely, pre-pandemic period) in 21 Brazilian hospitals. A pairwise analysis compared between the pre- and post-pandemic periods, with microbiologically confirmed central line-associated bloodstream infection (CLABSI) and ventilator-associated pneumonia (VAP) incidence density (cases per 1000 central line and ventilator days, respectively) presenting the proportion of organisms that caused healthcare associated infections (HAI) and antibiotic consumption.

#### 2.1.3. Gram-Positive Bacteria

Unlike Gram-negatives bacteria, Gram-positive bacteria have shown less global antimicrobial resistance (AMR) occurrence during and after the COVID-19 pandemic. In detail, global AMR prevalence in Gram-positive microorganisms was 19.86% in 2018, 13.56% in 2019, 18.12% in 2020, and 12.41% in 2021. Moreover, in comparing MRSA and VRE, no risk of association has been found between the COVID-19 pandemic and the incidence (IRR 0.99, 95% CI: 0.67–1.47, I2 = 91%, *n* = 8) or proportion (RR 0.91, 95% CI: 0.55–1.49, I2 = 92%, *n* = 12) of resistant Gram-positive cases [18,25]. The original data in the meta-analysis belong to a comparison study on CLABSIs, ventilator-associated events (VAEs), and catheter-associated urinary tract infections (CAUTIs) reported monthly from each facility of the Veterans Affairs, US, to a centralized database before the pandemic (February 2019 through January 2020) and during the pandemic (July 2020 through June 2021). The reported presence of IPAC or ASP interventions was not associated with a statistically significant difference in resistance rates (reporting IPAC/ASP: RR 0.59, 95% CI 0.15–2.42, I2 = 89%, *n* = 4; not reporting IPAC/ASP: RR: 1.15, 95% CI: 0.94–1.41, I2 = 89% *n* = 8, subgroup difference test *p* 0.36) [26,27]. The data also derive from a meta-analysis study. The original data analyzed belong to a case–control study which ran from 2017 to 2020 and monitored hospital discharges over a four-month period in St. Andrea Hospital, Rome, Italy, as well as a prospective surveillance study on Texas Children’s Hospital (TCH) campuses, Texas, US.

Specifically, over 6,848,357 patient days of follow-up were analyzed in a meta-analysis, and it was found that the COVID-19 pandemic was not associated with a change in the rate of incidence of methicillin-resistant *Staphylococcus Aureus* infections (IRR 1.03, 95% CI: 0.65–1.62, I2 = 95%, *n* = 5) [24,28]. Accordingly, the COVID-19 pandemic was not associated with a change in the rate of MRSA cases (RR 0.91, 95% CI: 0.60–1.36, I2 = 93%, *n* = 7) [27]. Further, a meta-analysis of the prevalence of vancomycin-resistant *Enterococci (VRE)* which considered over 356,056 patient days showed that the COVID-19 pandemic was not associated with a change in the incidence of VRE (IRR 0.75, 95% CI: 0.49–1.15, I2 = 56%, *n* = 3). This is concordant with the absence of a change in the proportion of VRE cases (RR 0.91, 95% CI: 0.30–2.79, I2 = 94%, *n* = 5) [27,29]. The data were derived from a meta-analysis assessment. The original data analyzed belong to a perspective study (namely, 2 April 2014 to 13 August 2020) conducted at Copenhagen University Hospital Bispebjerg, Denmark (see Table 1 for other studies reporting AMR during the 2020–2022 COVID-19 pandemic period).

#### 2.1.4. Fungi

Fungi are organisms that potentially affect immunocompromised patients [30]. However, during the last waves of pandemic and at its ending, several cases of pulmonary infection by fungi were also reported in non-immunocompromised subjects. Thus, they are an emerging issue in the post-pandemic phase of COVID-19 [29].

Fungi produce high amounts of enzymes, which are the main target of the of antifungal agents. Likewise, fungi are able to prevent the suppression of metabolic processes driven by such enzymes. Fungi also change the arrangement of a targeted enzyme, which significantly reduces the effectiveness of azole antifungals. Furthermore, fungal cells can actively pump out antifungal drugs via efflux pumps. Thus, they bypass the metabolic cascades targeted by drugs (e.g., inhibition of ergosterol biosynthesis at the C-14 demethylation stage, non-competitive inhibition of squalene epoxidase, the condensation of serine and fatty acyl-Coenzyme A catalyzed by serine palmitoyltransferase, plasma membrane H1 ATPase). Finally, fungi can produce and deliver extracellular enzymes capable of dissociating antifungal compounds [31].

In a study which ran during the COVID-19 pandemic outbreak, life-saving surgeries, early diagnosis, and targeted treatments for co-fungal infections were prevalent in the hospital-admitted population. In fact, Posteraro et al. have followed-up on the 53-day clinical course of a COVID-19 patient suffering from type 2 diabetes mellitus with bloodstream infection by three different organisms, namely *Morganella morganii*, MRSA, and *Candida glabrata* [32]. After 13 days of treatment with caspofungin, *C. glabrata* was found to have FKS-associated pan-echinocandin resistance. A recent systematic analysis has demonstrated that high-affinity iron uptake mechanisms are another critical virulence determinant in *C. glabrata AMR* [33]. Furthermore, retrospective reports conducted in New Delhi (India) described candidemia to be found in 15 critically ill COVID-19 patients. In 10 cases, *Candida auris* (multidrug resistant) MDR was responsible for six deaths [34]. Mohamed et al. have observed a severe COVID-19 pneumonia case with co-infection with *Aspergillus fumigatus*, a multi-triazole-resistant strain [35]. In a retrospective, single-center case series of 31 subjects, 19.4% were found to suffer from aspergillosis. In other studies from Indian hospitals, candidemia was detected in 2.5% of critically ill patients, with a mortality of 53%. Interestingly, 66% of the patients suffered from persistent fungaemia despite treatment with antifungal therapy [36]. In conclusion, we must recognize that data from the literature on FKS-associated pan-echinocandin resistance development are few and, fortunately, do not confirm the emergence of detrimental *Candida glabrata* AMR worldwide.

Although the azole group has established itself as the standard treatment against fungal infections, COVID-19 patients have contributed to a change in this paradigm. In fact, Denning et al. have demonstrated an increased occurrence of triazole resistance by *Aspergillus fumigatus* among patients with chronic fungal diseases in a randomized clinical trial (RCT) [37].
antibiotics-13-00233-t001_Table 1Table 1Main studies on antibiotic resistance developed during the COVID-19 pandemic (2020–2022).Type and Number of PatientsGerms’ ResistanceMain Antibiotic ResistanceReference340 outpatients/inpatients*E. coli*, *Klebsiella*, *S. aureus (MSSA)*, *S. aureus (MRSA)*, *P. aeruginosa., and Enterobacter species*Cotrimoxazole, piperacillin, ceftazidime, and cefepime[38]102 ICU patients*A. baumannii*, *K. pneumoniae*, *and S. maltophilia*Carbapenem and methicillin[39]190 ICU patients*K. pneumoniae*, *A. baumannii*, *S. maltophilia*, *C. albicans*, *and Pseudomonas* spp.Carbapenem[40]750 ICU patients*A. baumannii*, *and K. pneumonia*MDR, carbapenem[41]611 ICU patients*Acinetobacter* spp.Imipenem, meropenem, and ciprofloxacin[42]197 ICU patients*K. pneumoniae and A. baumannii*(PDR)*K. pneumoniae* and (MDR) *A. baumannii*[43]856 ICU patients*E. coli and K. pneumonia*Ciprofloxacin and ampicillin (*E. coli*); ampicillin and amoxycillin (*K. pneumoniae*)[44]255 outpatients/inpatients*S. aureus and P. aeruginosa*Oxacillin, vancomycin, carbapenems, colistin, third- and fourth-generation cephalosporins[45]7309 ICU patients*A. baumannii and E. coli*MDR[46]3532 outpatients/inpatients*E. coli, K. pneumoniae*, *and P. aeruginosa*ESBL producing Enterobacterales MDR[47]553 ICU patients*K. pneumonia and A. baumannii*Carbapenem resistant[48]Legend: MDR: multidrug resistant; PDR: pan-drug resistant; ICU: intensive care unit.

#### 2.1.5. Virus and Protozoa

Among protozoa, *Plasmodium falciparum* is a significant example of the impact of COVID-19 pandemic. The use of plants for healthcare is prevalent in low-income countries, such as those in Africa [49,50]. These remedies are preferred mainly because of their accessibility and low cost [50,51]. COVID-19 prevention has benefited from an increased use of these remedies. *Artemisia annua* is a plant containing artemisinin, and also used by African people to prevent malaria [52]. The use and abuse of the plant are endemic in tropical and subtropical low-income countries and have led to an increase in *Plasmodium* resistance to antimalarial drugs, including artemisinin and its derivatives. Therefore, such resistance will affect malaria control in these regions and jeopardize efforts to eliminate malaria by 2030 [15]. In the case of antimalarial drug resistance, it is dependent on the initial genetic event that produces the resistant mutant and, subsequently, on the selection of the survival having an advantage in the presence of the antimalarial drug [15].

There is a diversity of *Artemisia annua* species and, for the same species, the artemisinin content can significantly range from one region to another [15]. Thus, these *Artemisia* extracts can exert harmful drug pressure over a long period of time once their concentrations fall below the critical threshold to treat the parasites [53]. In Africa, resistance to chloroquine had led to its replacement by artemisinin-based combination therapies [53].

Enteroviruses and rhinoviruses were highly prevalent compared to other viruses during the pandemic, despite the preventive measures. In particular, data from a meta-analysis and systematic review of the literature reveal that there was an increase in the prevalence of non-SARS-CoV2 viruses in the second half of the pandemic (namely, from July 2021 to December 2022) and in the post-pandemic period also. Furthermore, non-COVID-19 patients showed a higher prevalence for influenza, seasonal coronaviruses, and human parainfluenza viruses [54]. The same meta-analysis and systematic literature review methods have shown that human respiratory syncytial virus undergoes a shift in incidence from autumn and winter season to spring [54].

The SARS-CoV2 pandemic has led to the use of several antiviral compounds: monoclonal antibodies, and the direct antiviral agents nirmatrelvir, ritonavir, molnupiravir, and remdesivir. The agents have been used for the early treatment of COVID-19. However, they have shown a very variable effectiveness, safety profile, and several limitations because of patients’ comorbidities and/or antiviral resistance. In detail, all neutralizing monoclonal antibodies, including tixagevimab–cilgavimab, have already been pulled-out according to the high proportion of resistance of multiple Omicron sub-variants. This is due to the fast rate of mutations in spike protein of the virus. On the other hand, monoclonal antibody combinations can target more conserved and non-overlapping epitopes on the SARS-CoV2 spike protein.

The current three direct antiviral agents (DAA)s (namely, nirmatrelvir, molnupiravir, and remdesivir) show varying in vitro potency against SARS-CoV-2 variants compared to the ancestral strains. In particular, remdesivir has been sporadically reported in a case series as encountering resistance among immunocompromised patients [55]. Furthermore, the mechanism of action of molnupiravir (namely, lethal mutagenesis/error catastrophe) offers high levels of transition/transversion mutation ratios in SARS-CoV-2 genomes. After 7 days of molnupiravir treatment, greater mutation rates vs. nirmatrelvir-ritonavir-treated patients were found. Interestingly, in this phase IIa clinical trial, there were no specific mutations conferring resistance to molnupiravir or nirmatrelvir [56].

### 2.2. Factors Involved in AMR Development during COVID-19

#### 2.2.1. Hospital Use of Antibiotics during the Pandemic

During the COVID-19 pandemic, there was a significant decrease in antibiotic prescriptions worldwide in both SARS-CoV2-infected and non-infected patients. On the other hand, a considerable increase in the use of antibiotics within hospitals has been recorded, especially in COVID-19 admitted patients. The latter can explain the initial significant increase in antibiotic resistance. Thus, strains such as broad-spectrum and anti-methicillin-resistant *Staphylococcus aureus* emerged. There was also an excessive use of antifungals (e.g., in intensive care departments) [57].

One retrospective study reported that colonization of carbapenem-resistant *Enterobacterales* increased from 6.7% to 50% in COVID-19 patients from 2019 to 2020 [15,16]. In China, Li et al. reported the isolation of 159 bacterial strains from 102 COVID-19 patients with secondary infections [16]. The most common was *Acinetobacter baumannii* (35.8%; *n* = 57), followed by *Klebsiella pneumoniae* and *Stenotrophomonas maltophilia* (30.8%; *n* = 49, and 6.3%; *n* = 10, respectively). Interestingly, *A. baumannii* and *K. pneumoniae* had a carbapenem-resistance rate of 91.2% and 75.5%, respectively. Another monocentric retrospective French study found that 26 COVID-19 patients admitted to an intensive care (ICU) with severe respiratory disease had bacterial co-infections (five isolates were resistant to third-generation cephalosporins and two to amoxicillin/clavulanate, respectively) [58]. Similarly, 19 ICU COVID-19 patients had co-infection by 17 MDR *A. baumannii* and one MRSA, according to retrospective data [17]. Further, Fu et al. have prospectively reported five cases of ICU COVID-19 patients coinfected by MDR germs (namely, extended-spectrum β-lactamase (ESBL)-producing *K. pneumoniae*, *Burkholderia cepacian*, *P. aeruginosa*, and *S. maltophilia*) [59]. In New York, US, New Delhi metallo-β-lactamase (NDM)-producing *Enterobacter cloacae* was the causative agent of secondary infections in five COVID-19 patients in a retrospective, single-site cohort study which aimed to validate a procalcitonin-guided algorithm to rationalize empirical antimicrobial prescriptions in non-critically ill patients with COVID-19 pneumonia [60].

A retrospective study on COVID-19 patients found 1959 rare isolates with 29% (569) having resistant pathogens. More specifically, MRSA, carbapenem-resistant *Acinetobacter baumannii*, *Pseudomonas aeruginosa*, *Klebsiella pneumoniae*, and *Candida auris* were the most commonly isolated organisms in a COVID-19-dedicated hospital according to a case series [61,62]. It is interesting to mention that this study also reported a higher prevalence of antimicrobial resistance outside of Europe.

The in-hospital antibiotic consumption rate is determined by defined daily doses (DDD) per number of bed days or patient days. From a critical point of view, changes in this measure do not necessarily reflect absolute changes in overall antibiotic consumption according to the absolute number of patients evaluated. As an example, the Veterans’ Health Administration data confirm an increase in antibiotics used per 1000 hospital days in 2020. However, overall antibiotic use was decreased, which was likely due to reductions in healthcare utilization related to non-COVID-19 conditions [63]. Accordingly, UK national surveillance data show that total antibiotic consumption (namely, DDD per 1000 inhabitants per day) decreased by 11% between 2019 and 2020. Indeed, considering DDD per 1000 hospitalizations, antibiotic use in hospitalized patients increased by nearly 5%. This shift reflects changes in the hospital population sustained by the COVID-19 pandemic. In fact, 72% of COVID-19 patients received antibiotic treatment, either on an empiric basis or to treat a confirmed bacterial co-infection [64]. It is important to note that several superinfections are due to hospital-acquired infections (namely, ventilator-associated pneumonia and central line-associated bloodstream infections). Consequently, in a multicenter observational cohort study, antibiotic use at 17 South Carolina hospitals showed a significant increase in those structures with COVID-19 patients [65].

The first waves of the SARS-CoV2 pandemic drained as many resources as possible from the healthcare armamentarium due to the suggestion that antibiotics could help improve the prognosis of patients. Indeed, high antibiotic administration was observed in hospitals during the pandemic. Realistically, this misuse could be a justifiable treatment only in case of secondary bacterial or fungal infections.

#### 2.2.2. Preventive Measures: PPE and Disinfectants

An increased use of PPE occurred during the COVID-19 pandemic within our communities and, especially, in the hospital setting. Thus, PPE use has contained SARS-CoV2 diffusion but has also contributed to an increase in the spread of MDROs. This is explained by the dramatic initial shortage of PPE in high-, middle- and low-income countries. In fact, MDROs were first been isolated from COVID-19 units. For example, the incidences of *Candida auris* and carbapenem-resistant organisms (CROs) were a common finding, probably due to wearing of the same PPE in the care of multiple patients, according to a retrospective Argentinian study [66]. A further associative factor has been the lack of healthcare personnel. Staffing shortages existed due to the following reasons: (1) healthcare workers’ sick leaves; (2) sick leaves due to conditions other than COVID-19 (e.g., burnout, other mental illnesses), often occurring during the pandemic; (3) healthcare workers’ quarantine following SARS-CoV2-positive household contacts; (4) and a relative lack of healthcare workers because of the increased need for care for COVID-19 patients. These reasons are stated according to a perspective cohort study as well as a retrospective analysis of Japanese data [67,68].

During the pandemic, there was an extensive use of biocides (namely, disinfectants, antiseptics, and/or preservatives) in both the community and hospital settings. In this regard, their impact on AMR, including cross-resistance to unrelated antimicrobials, is debated. In fact, biocides can lead to drug resistance through modification of a germ’s membrane; through over-regulation of efflux pumps; by increasing the propensity to form biofilms; or through the introduction of a virtual but non-cultivable state that allows for survival in adverse environments [69].

Outside of hospitals, the use of chemicals as a result of the excessive consumption of pharmaceuticals and disinfectants (e.g., quaternary ammonium compounds (QACs) and trihalomethanes (THMs)) has imposed an unprecedented selective pressure on AMR. Thus, forty environmental samples covering water and soil matrices from the surroundings of designated hospitals in Wuhan were collected in March 2020 and June 2020, respectively. The chemical concentrations and profiles of the antibiotic resistance genes (ARGs) were revealed. They were used in ultra-high-performance liquid chromatography, tandem mass spectrometry, and metagenomics studies. Interestingly, the selective pressure of pandemic-related chemicals increased 1.4–5.8 times in March 2020, and then decreased to the normal level of the pre-pandemic period in June 2020. Therefore, the relative abundance of ARGs under selective pressure growth was 20.1 times higher than under normal selection pressure. Furthermore, the effect of QACs and THMs in exacerbating the prevalence of antimicrobial resistance, as elaborated through null model, variation partitioning, and co-occurrence network analyses, showed that QACs and THMs have close interaction with efflux pump genes and mobile genetic elements, respectively. These two items contributed to more than 50% to the formation of the ARG profile. More specifically, QACs enhanced the cross-resistance carried out by qacEΔ1 and cmeB up to 3.0-fold more; THMs increased ARG horizon transfer by 7.9-fold to initiate microbial responses to oxidative stress. Moreover, quinolone efflux pumps encoding qepA and β-lactamases encoding oxa-20 were identified as priority ARGs with a potential risk to human health under upward selective pressure [70]. In addition, QACs and THMs synergistically promoted the spread of antimicrobial resistance to elicit cross-resistance and increase the spread of ARGs. Specifically, qepA and oxa-20 were selected as priority ARGs with a high probability of affecting human health. Indeed, the main genes resembling ARGs in the post-COVID-19 era were as follows: mutations in regulators (e.g., *acrR, marR, soxR, and crp*), outer membrane proteins and transporters (*mipA and sbmA*), and RNA polymerase genes (*rpoB and rpoC*) [71].

It is useful to mention that while we are no longer in a pandemic but are instead in an endemic phase of the SARS-CoV2 diffusion, disinfectants continue to be extensively used in hospitals and elevators [60].

Finally, the microbicides found in most disinfectants and surface cleaners usually end up in wastewater treatment plants and other bodies of water. Importantly, when the concentration of a biocide is very high, there can be maximum inhibition of the bacteria growth. However, when their concentration increases and does not reach the minimum inhibitory concentration, this can play in favor of antimicrobial resistance.

#### 2.2.3. Travel Restrictions and Re-Opening

The ever-increasing spread of MDROs is certainly facilitated by international travel. In fact, the first lockdown periods were followed by a lower spread of germs resistant to antibiotics.

Furthermore, upon the reopening of inter- and extra-continental travelling, AMR has become a worldwide issue. This phenomenon can be explained by low-/middle-income countries having a high prevalence of AMR, mostly of Gram-negative bacteria. Practically up to 80% of travelers to South Asia are at least temporarily colonized by ESBL-producing *Enterobacterales*. Once colonized, more than 10% remain colonized for at least one year. Subsequently, 12% will transmit the colonization to another family member, perpetuating the circulation of the MDRO.

From a genetic point of view, the primary gene responsible for resistance to carbapenems was initially identified in India. Further, the mcr1 gene, responsible for resistance to colistin, was initially isolated in China. Indeed, these genes are spread throughout the world as confirmed by the SENTRY Antimicrobial Surveillance Program data from 2014 and 2015 [72]. For the first time, the emergence of the tigecycline resistance gene tet (X4) was recently found in China in 2011 and 2014. Intriguingly, one of the AMR genes has been significantly linked to the pandemic, namely CTX-M [73].

The “reservoir” role of low-income countries (LICs) in AMR at the reopening of travel restrictions can also explain their role. Indeed, the use of antibiotics in outpatients during the pandemic showed a decrease in prescription within high-income countries (HICs) during the year 2020, according to the US IQVIA Total Patient Tracker [18,27,74]. For example, in the United States, the consumption of antibiotics decreased by 33%; in Canada decreased by 26%, based on data from IQVIA’s CompuScript database; and in Europe decreased by 18%, based on data from the European Surveillance of Antimicrobial Consumption Network [75,76]. These reductions can be explained by the lower spread of non-COVID-19 community diseases caused by the measures of social distancing which did not allow for human relationships and, consequently, prevented the spread of these diseases [77]. Furthermore, a decrease in influenza cases has also been registered in the pediatric population according to the annual epidemiological report of the European Centre for Disease Prevention and Control. In this regard, in Australia there was a significant decrease in respiratory syncytial virus infections [78,79]. Moreover, the Invasive Respiratory Infection Surveillance Initiative, a prospective analysis of surveillance data from 26 countries, showed an ongoing modest sustained reduction in invasive disease due to *Streptococcus pneumoniae*, *Haemophilus influenzae*, and *Neisseria meningitides* [55,80].

On the other hand, these favorable trends in the context of HICs were not observed within LICs. The latter showed a significant increase in prescription of non-pediatric antibiotics (mainly, azithromycin) during the epidemic wave of COVID-19. The data have been systematically reviewed by using the NVivo 12 software and were analyzed by using both inductive and deductive thematic analyses [81] (Figure 1).

The figure describes the entire AMR process before and after the lockdown ending, describing the causative factors involved. Antibiotic use and misuse, protective measures and use of disinfectants/biocides, and lockdown restrictions and subsequent reopening contributed (arrows) to the emergence of AMR. Both before and after lockdown, there was a decrease in healthcare operators’ awareness of AMR risk and a weakness in stewardship programs. Legend: AMR: antimicrobial resistance.

## 3. Materials and Methods

We conducted a search on PubMed and Medline for original articles, reviews, meta-analyses, and case series using the following keywords, their acronyms, and their associations: antibiotics, antimicrobial resistance, Severe Acute Respiratory Syndrome CoronaVirus 2 (SARS-CoV2), COVID-19 pandemic, personal protective equipment. The last Medline search was dated 30 November 2023.

In detail, we retrieved 380 contributions, after duplicates were removed. Further, we excluded 198 contributions because of the type of the paper (review, non-English publication, book chapter). We further excluded 102 contributions because they did not address the searched topic. Among the 81 contributions included, we considered data from the main healthcare systems in Italy, Europe, and the US and the mentioned date of access.

Out of 81 contributions, we included 16 reviews; 10 meta-analyses; data from 13 main national and supranational databases; and 42 original articles (namely, retrospective, perspective, multicentric, and international studies).

## 4. Conclusions and Future Perspectives

AMR remains a severe and emerging healthcare system problem that is exacerbated and spread worldwide. PPE use initially masked the dimensions and proportions of the problem during COVID-19. The decline in the COVID-19 pandemic and PPE use has led to AMR diffusion, firstly in hospitals. Thereafter, populations, especially older and immuno-compromised individuals, have shown a growing incidence of infection by well-known bacteria and also fungi. The first-line antibiotics that were used early during the SARS-CoV2 pandemic have been replaced by second- and third-line more specific molecules, increasing the cost of treatments for citizens.

The impact of the COVID-19 pandemic on AMR transmission and circulation worldwide has been bimodal. At its beginning, the COVID-19 pandemic slowed down the transmission of resistant (AMR) pathogens (e.g., Gram-negative germs, especially in hospital settings). Subsequently, the release of travel restrictions has exponentially allowed for the proliferation of microorganisms with old and new AMR characteristics.

Although there has been a growth in the incidence and prevalence of Gram-negative AMR, as well as in only certain Gram-positive bacteria, in the post-pandemic period, recent meta-analyses and systematic reviews have shown that the specific association with COVID-19 is lacking. On the other hand, the role of COVID-19 in secondary AMR infection outbreaks is more relevant due to the excessive use and misuse of antimicrobials in hospitalized patients during the COVID-19 period. The emergence of fungal microorganisms with multi-resistance in the post-COVID-19 period is still a matter of debate.

A specific and recognized role in the circulation of and increase in the diffusion of antibiotic resistance genes is reserved to the excessive use of biocides, especially in hospital settings.

Indeed, costs for the development of new treatments for multidrug-resistant germs have considerably grown, especially in the Western world. This has generated unprecedented stimuli for physicians and researchers seeking the solution, prevention, and management of this issue. An international anti-AMR program is anticipated, and a personalized patient approach is needed in the next decade.

Supranational antibiotic and antimicrobial stewardship programs are awaited and should be integrated with antimicrobial prescriptions according to the model of “personalized medicine”, perhaps benefiting from the large-database artificial intelligence analysis.

## Figures and Tables

**Figure 1 antibiotics-13-00233-f001:**
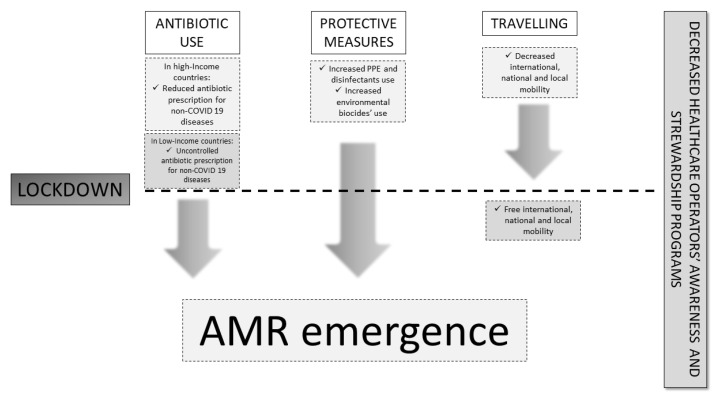
Factors influencing AMR development before and after the lockdown ending.

## Data Availability

All the data reviewed in this article are available online on the main medical databases (PubMed, Medline) and on the website of the National and International Microbiology, Internal Medicine, Infectious Disease meetings.

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
