# Peer review of "Post-Coronavirus Disease 2019 Pandemic Antimicrobial Resistance"

_antibiotics, 2024, doi:10.3390/antibiotics13030233_

Round 1

Reviewer 1 Report

Comments and Suggestions for Authors

In this review, the authors tried to find out the trends of antimicrobial resistance in the frame of the COVID-19 pandemic and post-pandemic era. Please find the following comments, which should be addressed before further processing:

General comments

·       The scientific name of any organisms should be in italics. Please correct it throughout the manuscript.

·       You tried to mention a “significant rise or increase”? What does that mean? How could you be sure that they significantly increased? Please clarify and correct if needed throughout the manuscript.

·       Please use the full form of any abbreviations when you use them for the first time. Please correct it throughout the manuscript.

·       There are a lot of published review articles on AMR and COVID-19. So, what was the novelty of your review? Please mention it properly in the introduction section.

Title

“SARS-CoV-2 POST-PANDEMIC” or “COVID-19 POST-PANDEMIC?” Also, please don’t use abbreviations in the title. SARS-CoV-2 is not a disease; it is a virus. Please correct it throughout the manuscript

Abstract

Line 20: Please provide the full form of COVID-19 here.

Line 22: “pandemic and post-pandemic era”. But you mentioned only post-pandemic in the tile.

Line 25 and 37: “antibiotics resistance” should be “antibiotic resistance.” Also, provide the full form of SARS-CoV-2 in Line 25.

Line 26: How were you sure to say “significant rise?” Did you perform any analyses? The same is in Line 33. Please clarify.

Line 37-38: “personal protection equipment” should be “personal protective equipment.” Please correct it throughout the manuscript.

Introduction

Line 61: “Thus, the aim of this review is to….” should be “Thus, the aim of this review was to…”. Also, “define the concept of antibiotic resistance” – what does that mean? Could you find any new concepts for antibiotic resistance? Please clarify.

Line 62: “before, during and after”, but the title doesn’t say that. Please clarify. Also, “We will review….” should be “We have reviewed…..”

Line 63-65: This should be in the conclusion section.

Results

Only “Results” or “Results and Discussions"?

Line 68-83: I can’t understand why you put it in the result section. Is this concept new to the scientific community? Please clarify.

Line 89: Why have you mentioned “Candida glabrata” under the Gram-negative bacteria subsection? Please clarify this.

Line 91-138: I can’t understand this portion of the results section properly. Did the authors perform any analyses to get these results?

Line 223, 276: “3.1.3” should be “3.1.2”; “3.1.4” should be “3.1.3”.

M+M

This section is very poor. Please improve it by providing more information, e.g., how you extracted information? Who extracted that information? Did you follow any review protocol to conduct this study? Did you select any criteria to include or exclude data or articles? Did you focus only on online open-accessed articles or also close-accessed articles? If so, how did you access them, or how did you download those articles? How many times did you check your extracted articles or data? How did you avoid duplication? How many articles did you select for this review?

Also, I found some statistical analysis data in the result section. Did you perform those analyses? If yes, how did you do that? You should mention it. If you didn’t, what was your credit? Please clarify this.

Conclusions

It seems fine to me.

Author Response

REVIEWER 1:

In this review, the authors tried to find out the trends of antimicrobial resistance in the frame of the COVID-19 pandemic and post-pandemic era. Please find the following comments, which should be addressed before further processing:

General comments

  • The scientific name of any organisms should be in italics. Please correct it throughout the manuscript.

We have provided it.

  • You tried to mention a “significant rise or increase”? What does that mean? How could you be sure that they significantly increased? Please clarify and correct if needed throughout the manuscript.

We thank the reviewer for the observation. We have checked and corrected the use throughout the manuscript.

  • Please use the full form of any abbreviations when you use them for the first time. Please correct it throughout the manuscript.

We have checked and corrected the use throughout the manuscript.

  • There are a lot of published review articles on AMR and COVID-19. So, what was the novelty of your review? Please mention it properly in the introduction section.

We thank the reviewer for the comment. We have made it.

Title

“SARS-CoV-2 POST-PANDEMIC” or “COVID-19 POST-PANDEMIC?” Also, please don’t use abbreviations in the title. SARS-CoV-2 is not a disease; it is a virus. Please correct it throughout the manuscript.

We thank the reviewer for the comment. We have changed the title.

Abstract

Line 20: Please provide the full form of COVID-19 here.

It has been integrated.

Line 22: “pandemic and post-pandemic era”. But you mentioned only post-pandemic in the tile.

We made the change.

Line 25 and 37: “antibiotics resistance” should be “antibiotic resistance.” Also, provide the full form of SARS-CoV-2 in Line 25.

We have made the changes and adding.

Line 26: How were you sure to say “significant rise?” Did you perform any analyses? The same is in Line 33. Please clarify.

We have specified the increase of incidence, according to the reviewed data from literature. In particular, we used metanalysis and systematic reviews data.

Line 37-38: “personal protection equipment” should be “personal protective equipment.” Please correct it throughout the manuscript.

It has been replaced throughout the manuscript.

Introduction

Line 61: “Thus, the aim of this review is to….” should be “Thus, the aim of this review was to…”. Also, “define the concept of antibiotic resistance” – what does that mean? Could you find any new concepts for antibiotic resistance? Please clarify.

We have made the changes. We thank the reviewer for the annotations.

Line 62: “before, during and after”, but the title doesn’t say that. Please clarify. Also, “We will review….” should be “We have reviewed…..”.

We thank for the observations. We have made the changes.

Line 63-65: This should be in the conclusion section.

We agree with the reviewer. We have deleted the sentence.

Results

Only “Results” or “Results and Discussions"?

We thank for the useful suggestion. We have made the change.

Line 68-83: I can’t understand why you put it in the result section. Is this concept new to the scientific community? Please clarify.

We thank the reviewer for the comment. Indeed, the definition of AMR is a result, as suggested in the Aims sentence of the Introduction section.

Line 89: Why have you mentioned “Candida glabrata” under the Gram-negative bacteria subsection? Please clarify this.

We agree and have removed these.

Line 91-138: I can’t understand this portion of the results section properly. Did the authors perform any analyses to get these results?

This is an interesting observation. We have extrapolated analysis data from review, original studies and metanalysis belonging to literature.

Line 223, 276: “3.1.3” should be “3.1.2”; “3.1.4” should be “3.1.3”.

We agree and we have corrected these.

M+M

This section is very poor. Please improve it by providing more information, e.g., how you extracted information? Who extracted that information? Did you follow any review protocol to conduct this study? Did you select any criteria to include or exclude data or articles? Did you focus only on online open-accessed articles or also close-accessed articles? If so, how did you access them, or how did you download those articles? How many times did you check your extracted articles or data? How did you avoid duplication? How many articles did you select for this review?

We have updated the Methods. Indeed, this is not a systematic review but we have updated more informations.

Also, I found some statistical analysis data in the result section. Did you perform those analyses? If yes, how did you do that? You should mention it. If you didn’t, what was your credit? Please clarify this.

We have updated the Methods’ section.

The statistical data have been extrapolated from metanalysis, reviews cited in this manuscript.

Conclusions

It seems fine to me.

Thank for the appreciation.

Reviewer 2 Report

Comments and Suggestions for Authors

The manuscript entitled "ANTIMICROBIAL RESISTANCE AT THE TIME OF SARS-CoV-2 POST-PANDEMIC" provides a comprehensive review of the use of antibiotics during the post-pandemic era and the development of AMR during this period. 

The authors highlight the significant issue and provide an in-depth review on the development, emergence, and prevalence of AMR reports in various organisms including bacteria and fungi. However, I suggest adding data regarding antiviral drugs as these drugs have also been in use during the era of the Covid-19 pandemic.  

Furthermore, data regarding antiprotozoal could be beneficial. 

Comments on the Quality of English Language

The English language is good. 

Author Response

REVIEWER 2:

The manuscript entitled "ANTIMICROBIAL RESISTANCE AT THE TIME OF SARS-CoV-2 POST-PANDEMIC" provides a comprehensive review of the use of antibiotics during the post-pandemic era and the development of AMR during this period. 

The authors highlight the significant issue and provide an in-depth review on the development, emergence, and prevalence of AMR reports in various organisms including bacteria and fungi. However, I suggest adding data regarding antiviral drugs as these drugs have also been in use during the era of the Covid-19 pandemic. 

We thank the reviewer for the suggestion. We have added the data. 

Furthermore, data regarding antiprotozoal could be beneficial. 

We thank the reviewer for the suggestion. We have added the data. 

Reviewer 3 Report

Comments and Suggestions for Authors

First of all, I would like to congratulate the authors for the innovative idea to analyse antibiotic resistance in relation with COVID pandemic. The following lines aims to remind the natural flow of ideas within an article.  An idea of the article should be formulated as "the aim". How we test the theory/hypothesis, how we select studies/patient/germs, what statistical analysis we use, when was the study conducted is described in the Material and Methods section. The results describe facts the confirm/infirm our hypothesis, while The discussion section talks about comparisons, interpretation of statistical analysis. Introduction states the current knowledge about the subject/theme/hypothesis. The conclusion (2-3 ideas) comes as an argument to confirm or infirm the hypothesis.

My concerns regarding the article are as follows:

1.The introduction is very general, worldwide ideas about COVID (there is a lot to be said but the aim is to describe the current state of antibiotic resistance). Please rearrange ideas and develop some new ones, more specific to the theme.

2. The aim of the study as described in introduction section "is to define the concept of antibiotic resistance, its incidence and characteristics before, during and after the COVID pandemic". There is no comparison between bacterial antibiotic resistance regarding a specific period of time. Please add details or modify the aim of the article accordingly.

3. Materials and methods section should be more descriptive and contain much more details regarding the selection of articles, type of germs, period of time and cultures included in the study. Plese add details.

4. Both the data about gram negative bacteria and gram-positive bacteria show no increase incidence of antibiotic resistance, meanwhile you conclude that there is an "growing incidence of infections in...." (line 324). the first part of the articles talks about antibiotic resistance and the conclusion is about infections. Please correct or add supplementary information.

5. Subsection 2.1.4 Fungi. The first two paragraphs I believe they should be in the Introduction section. Please add details/comparisons between antimicrobial resistance before, during and after COVID pandemic.

6. Chapter 3.1.1. Lines 181-201 should be merged with Chapter 2.1.2/2.1.3 and also interpreted accordingly. 

7. Please revise the Conclusions. The main ideas should refer to main/most important ideas within the article.

Author Response

REVIEWER 3:

First of all, I would like to congratulate the authors for the innovative idea to analyse antibiotic resistance in relation with COVID pandemic. The following lines aims to remind the natural flow of ideas within an article.  An idea of the article should be formulated as "the aim". How we test the theory/hypothesis, how we select studies/patient/germs, what statistical analysis we use, when was the study conducted is described in the Material and Methods section. The resultsdescribe facts the confirm/infirm our hypothesis, while The discussion section talks about comparisons, interpretation of statistical analysis. Introduction states the current knowledge about the subject/theme/hypothesis. The conclusion (2-3 ideas) comes as an argument to confirm or infirm the hypothesis.

My concerns regarding the article are as follows:

1.The introduction is very general, worldwide ideas about COVID (there is a lot to be said but the aim is to describe the current state of antibiotic resistance). Please rearrange ideas and develop some new ones, more specific to the theme.

We thank the reviewer for the suggestions. We have updated the Introduction.

  1. The aim of the study as described in introduction section "is to define the concept of antibiotic resistance, its incidence and characteristicsbefore, during and after the COVID pandemic". There is no comparison between bacterial antibiotic resistance regarding a specific period of time. Please add details or modify the aim of the article accordingly.

We thank the reviewer for the observations. We have changed and revised the sentence.

  1. Materials and methods section should be more descriptive and contain much more details regarding the selection of articles, type of germs, period of time and cultures included in the study. Plese add details.

We have added more particulars.

  1. Both the data about gram negative bacteria and gram-positive bacteria show no increase incidence of antibiotic resistance, meanwhile you conclude that there is an "growing incidence of infections in...." (line 324). the first part of the articles talks about antibiotic resistance and the conclusion is about infections. Please correct or add supplementary information.
  2. Subsection 2.1.4 Fungi. The first two paragraphs I believe they should be in the Introduction section. Please add details/comparisons between antimicrobial resistance before, during and after COVID pandemic.

We thank the reviewer for the observation. We have added data on Fungi to the Introduction.

  1. Chapter 3.1.1. Lines 181-201 should be merged with Chapter 2.1.2/2.1.3 and also interpreted accordingly. 
  2. Please revise the Conclusions. The main ideas should refer to main/most important ideas within the article.

We have revised the Conclusions.

Round 2

Reviewer 1 Report

Comments and Suggestions for Authors

Thanks to the authors for addressing my comments. However, I have a few more comments as follows:

Title: Please don't use abbreviations in the tile.

Line 108-110: Reference?

Line 202: Virus and Protozoa? What does that mean? Only Protozoa?

"Please make italics for all the genes' names, e.g., 334-336 lines.

Author Response

REPLY TO REFEREES’ COMMENTS:

REVIEWER 1:

Thanks to the authors for addressing my comments. However, I have a few more comments as follows:

Title: Please don't use abbreviations in the tile.

Ok. We have modified these.

Line 108-110: Reference?

Thanks, updated.

Line 202: Virus and Protozoa? What does that mean? Only Protozoa?

We thank the reviewer for the observation.  Thus, we have updated knowledge about virus prevalence/incidence during and after the pandemic, integrating the evidence on antivirals’ emergence of resistance.

"Please make italics for all the genes' names, e.g., 334-336 lines.

We have corrected these.

Reviewer 2 Report

Comments and Suggestions for Authors

Dear authors, 

Thank you for the revision of manuscript. 

Comments on the Quality of English Language

English language is OK.

Author Response

REVIEWEER 2:

Dear authors, 

Thank you for the revision of manuscript. 

We thank the reviewer for the appreciation.

Reviewer 3 Report

Comments and Suggestions for Authors

I have seen the added information within the article which bring improvement to the article.

Author Response

REVIEWER 3:

I have seen the added information within the article which bring improvement to the article.

We thank the reviewer for the appreciation.